# Glucose Homeostasis, Diabetes Mellitus, and Gender-Affirming Treatment

**DOI:** 10.3390/biomedicines11030670

**Published:** 2023-02-22

**Authors:** Charalampos Milionis, Ioannis Ilias, Evangelia Venaki, Eftychia Koukkou

**Affiliations:** Department of Endocrinology, Diabetes and Metabolism, Elena Venizelou General and Maternity Hospital, GR-11521 Athens, Greece

**Keywords:** glycemia, type 2 diabetes, insulin resistance, pathophysiology, transgender persons, testosterone, estrogen

## Abstract

The transgender (trans) population includes individuals with gender identities more fittingly aligned with the opposite sex or with an alternative that transcends the classical dipole of male/female. Hormonal treatment in transgender individuals aims to suppress the secretion of endogenous sex steroids and replace them with the steroids of the desired gender. The mainstay of gender-affirming treatment in transgender males is testosterone, whereas for transgender females it is estrogen, usually combined with an anti-androgen or a gonadotropin-releasing hormone agonist if testes are present. Testosterone and estrogen are involved in carbohydrate metabolism via direct effects on skeletal muscle, liver, adipose tissue, and immune cells and indirectly through changes in body fat mass and distribution. The effect of transgender treatment on glucose tolerance is not clear. The provided conflicting results demonstrate a positive, neutral, or even negative association between exogenous testosterone and insulin sensitivity in trans men. Studies show that feminizing hormonal therapy of trans women has mainly an aggravating effect on insulin sensitivity. The existing evidence is not robust and further research is needed to investigate the relationships between body fat distributions, muscle mass, and glycemia/insulin resistance in transgender people under hormonal therapy.

## 1. Background

### 1.1. Physiology of Insulin

Insulin is synthesized by the pancreatic beta-cells within the islets of Langerhans. It consists of two polypeptide chains, namely A (21 amino acids) and B (30 amino acids) which are connected via disulfide bonds. In the precursory molecule, the A and B chains are connected through the C-peptide. The release of insulin in the fasting state (basal secretion) follows a pulsatile pattern, with small secretory bursts occurring every few minutes. Postprandial release of insulin is biphasic, with an initial rapid secretion, followed by a less intense but more sustained hormonal release. The first phase reflects the release of already synthesized and stored hormone, whilst the second phase represents the secretion of both stored and newly produced insulin [1].

Blood glucose is the principal modulator of insulin release. The phosphorylation of glucose by glucokinase is the rate-limiting step of glucose metabolism that subsequently determines the insulin secretory response. Other factors, including amino acids, neural stimuli, other hormones, and medications, influence (stimulate, enhance, or inhibit) insulin synthesis and secretion [2]. After secretion, insulin enters the bloodstream and reaches peripheral tissues. Liver, muscle, and adipose tissue are the major targets of circulating insulin. The physiological role of insulin generally contains two arms: the metabolic and the mitogenic pathway. The metabolic effects are fundamental in regulating carbohydrate, lipid, and protein metabolism, whilst the mitogenic actions participate in promoting cell division and growth [3].

Insulin action is critical for the cellular uptake of glucose in insulin-dependent tissues. Within hepatocytes, insulin stimulates the utilization of glucose for synthesis and storage of glycogen and the conversion of excess glucose into fatty acids. It also downregulates gluconeogenesis and glycogenolysis in the liver either directly or indirectly through the paracrine suppression of pancreatic glucagon. Furthermore, insulin induces important effects in the skeletal muscle tissue. Specifically, it facilitates protein synthesis and promotes the conversion of glucose to glycogen for subsequent coverage of muscle energy expenditure. In the adipose tissue, insulin stimulates the synthesis and storage of triglycerides, which serve long-term energy needs [3,4]. Insulin not only lowers blood glucose and regulates nutrient metabolism, but it also directly modulates inflammatory mediators and acts upon immune cells to enhance immunocompetence. In this sense, hyperglycemia is a pro-inflammatory state, whilst insulin acts as an anti-inflammatory agent [5]. Other significant, but less studied, physiological roles of this hormone include, inter alia, actions related to endothelial function, neuronal plasticity and cognitive processes, homeostatic kidney functions, bone formation, and skin integrity. However, the precise responsible mechanisms remain elusive.

### 1.2. Blood Glucose Regulation and Insulin Resistance

Glucose homeostasis reflects a balance between dietary carbohydrate intake and uptake and utilization of glucose by peripheral tissues. The pancreatic secretion of insulin is the most important regulator of glycemia. In the fasting state, low insulin levels promote glucose production through hepatic gluconeogenesis and glycogenolysis and hinder glucose uptake in insulin-sensitive tissues (skeletal muscle and fat) in order to maintain euglycemia. Glucagon is secreted by pancreatic alpha-cells when blood glucose or insulin levels are low or during exercise, and it stimulates glycogenolysis and gluconeogenesis by the liver, and to a lesser extent, by the kidneys. Postprandial glucose absorption causes an increase in insulin and a decrease in glucagon, leading to the opposite processes. Insulin promotes the uptake of glucose by peripheral tissues. Most of the postprandial glucose is used by skeletal muscle [6]. Some tissues, mainly the brain, utilize glucose without the mediation of insulin. Further factors, such as neural stimulation, metabolic signals, and other hormones, also play a role in glucose metabolism.

Insulin resistance is a state of attenuated biological response of tissues to insulin action. In the setting of adequate islet function, pancreatic insulin secretion increases to maintain blood glucose within normal levels. Elevated endogenous insulin results in weight gain, which in turn aggravates insulin resistance, formulating a pathophysiological vicious cycle. The excess of insulin may be combined with a cluster of abnormalities, including dyslipidemia, hypertension, obesity, prothrombic state, endothelial dysfunction, inflammatory conditions, abnormal uric acid metabolism, and sleep apnea. Furthermore, the progressive inability of beta-cells to preserve the overproduction of insulin triggers the clinical onset of glucose intolerance and type 2 diabetes mellitus [1,3].

The causes of insulin resistance are complex, whilst the responsible biological mechanisms have not been entirely clarified. In the absence of autoimmunity or heritable syndromes, the excess of body weight is the major pathogenetic determinant of impaired sensitivity to insulin. Excessive nutrition may result in chronic inflammatory conditions, changes in lipid metabolism, and alterations of the gastrointestinal microbiota, which in turn can impede insulin action [7]. Moreover, a wide range of pharmacological agents—each one with different mechanism—may contribute to glucose intolerance. Thiazide diuretics, beta-blockers, sympathomimetics, corticosteroids, and sex hormones are commonly prescribed drugs which may have adverse effects on carbohydrate metabolism [8].

### 1.3. Diabetes Mellitus

Diabetes mellitus consists of a group of metabolic disorders that manifest hyperglycemia as a common feature. The various types of diabetes mellitus are pathogenically related to the combined effect of genetic and environmental factors. There are two main categories of this disease based on the underlying pathophysiological processes that lead to the progression of hyperglycemia. Type 1 diabetes mellitus is caused by absolute insulin deficiency due to autoimmunity against pancreatic beta-cell components. Type 2 diabetes mellitus is characterized by varying degrees of insulin resistance, impaired insulin secretion, and increased hepatic glucose production. Other types of diabetes mellitus include clinical entities that are developed due to genetic defects in insulin secretion or action, abnormalities that impede insulin secretion, mitochondrial impairments, and conditions that impair glucose tolerance [9].

The metabolic dysregulation caused by chronic hyperglycemia is associated with potential secondary grave complications in multiple tissues and organs. Diabetes mellitus is a major cause of microvasculopathy, affecting the capillary blood vessels in the glomeruli, retina, myocardium, skin, and muscle. In addition, it appears to contribute to the causation of macrovasculopathy [10]. Diabetic complications may have an adverse effect on the individual’s quality of life and significantly burden the health care system. The current epidemiological situation of diabetes mellitus, along with its future prospects, is an important public health concern. Indeed, the global prevalence of diabetes mellitus has risen over the past four decades, and it is expected to grow further in the next years. Almost half a billion people worldwide are currently living with diabetes mellitus and their number is estimated to increase by 25% in 2030 and 51% in 2045 [11].

Prediabetes (in the form of impaired fasting glucose and/or impaired glucose tolerance) is a high-risk state for prospective development of overt diabetes. It is defined by intermediate glycemic values that are higher than normal, but lower than the diagnostic thresholds of diabetes mellitus. Irrespective of the underlying etiology, prediabetes is characterized by a pathophysiological defect in insulin action [12]. Reversion of prediabetes to normal glucose homeostasis is associated with significantly reduced chances of future hyperglycemia [13]. Hence, clinical practice should also focus on attainment and maintenance of both euglycemia and normal insulin sensitivity in case of interventions that could potentially jeopardize glucose metabolism.

## 2. Caring for Transgender People

### 2.1. Sex, Gender Identity, and Transgender People

In most mammals, including humans, the biological sex is determined by the makeup of sex chromosomes X and Y. The presence of an intact sex-determining region in the Y chromosome (SRY gene) drives the embryonic development of male sex [14]. In most cases, sexual dimorphism can be objectively defined through the examination of karyotype, internal and external genitalia, and secondary sex characteristics. In contrast, gender identity is a person’s subjective feeling of belonging (or not) to a certain type of gender, such as male, female, a combination of both, or neither, and it is not readily observable by others. It is not certain whether gender identity is a socially constructed concept, or it is stipulated by fundamental biological properties. Perhaps both factors play a role in the formation of gender identity, but the extent of each one’s contribution is unknown.

Gender dysphoria refers to the sense of distress which derives from the incongruence between the self-perceived gender identity and the biological sex (usually assigned at birth). It is an adverse emotional state that can interfere with social activities and other functional fields of life. Transgender (trans) people are individuals who have gender identities more fittingly aligned with the opposite sex or with an alternative that transcends the classical dipole of male/female [15]. In particular, transgender men are individuals with a male gender identity although their natal sex was female and transgender women are individuals who self-identify as females despite having been labeled as males at birth. The ‘transgender’ term also refers to people whose gender identity matches neither the male nor the female traditional role, such as agender, bigender, pangender, or gender-fluid humans [16].

Cultural norms have changed over time and gender diversity has gradually become more acceptable within society. Nowadays, the number of transgender people has significantly grown. However, it is not certain whether this increase reflects an actual trend, or it is a result of greater social acceptance and wider availability of therapeutic options, which encourage trans people to disclose their gender identity [17]. It is difficult to precisely assess the prevalence of the transgender population because of the heterogeneity among the relevant epidemiological studies. It is estimated that approximately 0.355% of the population identify themselves as transgender. Nevertheless, the prevalence of trans people who receive hormonal or surgical treatment for gender transition is only about 0.009% [18]. Although male-to-female gender transition was traditionally considered to be more frequent than female-to-male, there is a steady increase in the number of individuals seeking affirmation to the male gender over the past years. This tendency has begun to progressively close the gap in frequency [19].

### 2.2. Medical Treatment for Gender Transition

After the diagnosis of gender dysphoria is established and the desire for gender transition is confirmed, medical interventions can be considered for this purpose. Hormonal treatment in transgender individuals aims to suppress the secretion of endogenous sex steroids and replace them with hormones of the desired gender. The therapeutic goal is to maintain blood levels of sex hormones within the normal range of cisgender counterparts. The mainstay of gender-affirming treatment in transgender males is testosterone, routinely administered parenterally or transdermally. Typical regimens in transgender females consist of an oral or transdermal estrogen preparation, usually combined with an anti-androgen or a gonadotropin-releasing hormone agonist if testes are present. In non-binary persons, the treatment is modified according to the clinical targets [20].

Hormonal therapy in transgender people aims to induce physical changes that mimic the phenotype of the desired gender. The expected masculinizing results in trans men include development of thicker facial and body hair, loss of scalp hair, skin oiliness, muscular growth, reduced fat mass, deepening of the voice, cessation of menses, clitoral enlargement, and vaginal atrophy. Feminization in trans women is expressed with thinning of terminal hair, increased body fat, decreased muscle mass, breast growth, softening of the skin, diminished sexual desire and spontaneous erections, and testicular and prostate atrophy. Most of the changes start to appear within the first few months of gender-affirming treatment and maximum effects are achieved later, up two to five years [21].

Hormonal treatment is efficient and safe in general [22] and it improves the quality of life and psychological status of trans people [23]. Nevertheless, exogenous administration of sex hormones has been associated with certain metabolic side effects, although the existing evidence is not strong. Most concerns in transgender males are related to the potential occurrence of polycythemia, liver toxicity, and dyslipidemia with the use of testosterone. Hormonal therapy for transgender females entails possible risks of thromboembolic disease, liver dysfunction, hypertension, and hypertriglyceridemia. Hence, regular clinical evaluation and laboratory monitoring are recommended [21,24]. There is a theoretical possibility that cross-sex treatment could induce hormone-sensitive malignancies, especially after an extended period of use. Nonetheless, current evidence suggests that neither masculinizing nor feminizing therapy are associated with increased risk for carcinogenesis [25].

## 3. Sex-Specific Differences in Glucose Metabolism

Glucose homeostasis is not regulated identically in males and females. Knowledge of differences in glucose metabolism between sexes is useful when considering administration of any type of hormonal therapy with sex steroids. Body composition and fat distribution are closely associated with insulin sensitivity in humans. Adult men have greater lean tissue mass than women, probably because of the anabolic effects of testicular androgens. Although adult women have a higher percentage of fat mass, there is an increased visceral fat deposition in men. Consequently, there is greater glucose effectiveness and insulin sensitivity among females in comparison with males [26].

Due to a series of possible physiological properties, euglycemic women are more sensitive to insulin than men when matched for age, body mass index, and physical fitness. Firstly, both the proportion of type I muscle fibers and the capillary density are higher in women in comparison to men, favoring thereby a greater glucose uptake by skeletal muscle. Secondly, females exhibit both a greater capacity of insulin secretion and a stronger incretin effect in contrast to male counterparts [27]. Thirdly, sex is an important factor affecting microflora in the human gut. The male-specific intestinal microbiome may be partially responsible for the occurrence of glucose intolerance [28]. Of course, several other mostly unknown physiological processes might also be plausible causes. In general, sex-specific attributes in glucose homeostasis could play a significant role in shaping an optimal approach to diabetes prevention and management.

Sex-specific differences in glucose metabolism are possibly mediated not only by the action of gonadal steroids, but also by chromosomal factors genetically inherent to each sex [29]. Indeed, the influence of the genetic sex may generate differences in cellular function between XX females and XY males. In animal models, XX individuals exhibit increased fat deposits and food intake, and subsequently, a greater risk of insulin resistance resulting from a high-fat diet compared to XY counterparts, independently of the presence of the SRY gene, the type of gonads, and the circulating sex hormones. Perhaps the number of X chromosomes or holandric genes (other than the SRY) contribute to these sex differences [30]. Therefore, not all sex differences are modifiable by establishing a new profile of circulating sex steroids.

## 4. The Impact of Gender-Affirming Treatment on Glycemia

Insulin resistance is defined by an inability of circulating insulin to effectively regulate the uptake of glucose by target tissues. Initially, a compensatory hyperinsulinemia occurs to maintain normal blood glucose levels. After a period of elevated insulin requirements, pancreatic beta-cells may start to fail, consequently leading to non-physiological hyperglycemia and eventually to diabetes [31]. Due to conflicting research data, it is debatable whether gender-affirming hormonal therapy has an impact on this process or not. On one hand, there is evidence indicating that both trans men and women suffer more often from diabetes mellitus [32], whilst on the other hand, it has also been reported that the standardized incidence ratio is not elevated in transgender people compared with the general population [33,34]. Therefore, there is no clear association between hyperglycemia with cross-sex medical treatment and the possible underlying mechanisms are not fully understood.

Both testosterone and estrogen are involved in the carbohydrate metabolism via direct effects on skeletal muscle, liver, adipose tissue, and immune cells and indirectly through changes in body fat mass and distribution [35,36]. Insulin resistance has a central role in the pathogenesis of type 2 diabetes mellitus, but it is also associated with a variety of poor metabolic outcomes. Some indicated examples may include hypertension, dyslipidemia, obesity, and atherosclerotic cardiovascular disease, even in individuals without diabetes [37]. Thus, understanding the link—if any—between cross-sex hormonal treatment and insulin resistance may provide insights regarding possible health risks among the trans population. However, the existing evidence from relevant studies is limited and suffers from small sample sizes, short follow-up periods, and young age cohorts of participants [38]. Table 1 and Table 2 present findings from studies that investigated changes in insulin resistance after the initiation of gender-affirming treatment.

### 4.1. Glycemia in Transgender Males

Gender-affirming therapy in transgender men imitates the replacement treatment in hypogonadal cisgender males by administering testosterone in various formulations. The clinical purpose is to induce and maintain secondary masculine sex characteristics. The effect of treatment on glucose tolerance is not clear. There have been conflicting results demonstrating a positive [39,40], neutral [41,42], or even negative [43] correlation between exogenous testosterone and insulin sensitivity in trans men. Furthermore, it is not known whether androgens have a direct androgenic impact on glucose metabolism, or their action is due to changes in body mass index, weight, and visceral fat.

**Table 1 biomedicines-11-00670-t001:** Studies investigating insulin resistance in transgender males.

Study	Method	N	Duration	Insulin Resistance			
				Fasting Glucose	Fasting Insulin	HOMA-IR	Hyperinsulinemic Euglycemic Clamp
Shadid et al., 2020	Prospective (crossover control)	35	1 year	↔	↓	↓	
Auer et al., 2018	Prospective (crossover control)	24	1 year	↔	↓	↓	
Bretherton et al., 2021	Prospective (compared with cisgender women)	43	1 year			↔	
Elbers et al., 2003	Prospective (crossover control)	17	1 year	↓	↔		↔
Polderman et al., 1994	Prospective (crossover control)	13	4 months				↑

↓: decrease, ↔: no change, ↑: increase.

### 4.2. Glycemia in Transgender Females

For their gender transition, trans women follow an estrogen treatment. The latter is combined with either anti-androgen therapy or gonadotropin suppression unless orchiectomy has been performed. Their situation is comparable to that of menopausal women under hormonal replacement treatment. The existing evidence shows that feminizing hormonal therapy of trans women has a mainly aggravating effect on insulin sensitivity [39,40,41,42,43]. Nevertheless, no safe conclusions can be drawn upon the impact of exogenous treatment with estradiol on glucose homeostasis in transgender females due to the scarcity of the relevant data.

**Table 2 biomedicines-11-00670-t002:** Studies investigating insulin resistance in transgender females.

Study	Method	N	Duration	Insulin Resistance			
				Fasting Glucose	Fasting Insulin	HOMA-IR	Hyperinsulinemic euglycemic clamp
Shadid et al., 2020	Prospective (crossover control)	55	1 year	↔	↑	↑	
Auer et al., 2018	Prospective (crossover control)	45	1 year	↔	↑	↑	
Bretherton et al., 2021	Prospective (compared with cisgender men)	41	1 year			↑	
Elbers et al., 2003	Prospective (crossover control)	20	1 year	↔	↑		↑
Polderman et al., 1994	Prospective (crossover control)	18	4 months				↑

↔: no change, ↑: increase.

### 4.3. Sex Steroids and Insulin Resistance

The effect of gender-affirming treatment on glucose homeostasis is not completely understood. Hormonal therapy in transgender individuals affects body weight as well as lean and fat mass. Testosterone treatment results in a decrease in fat mass and an increase in lean mass in trans men, whilst estrogen and anti-androgen therapy have the reverse effects in trans women [44]. Gonadal steroids act on adipocytes through separate (androgen and estrogen) receptors. Subsequently, intracellular signaling influences the production of microRNAs, lipolysis and lipogenesis, hormonal secretion of adipocytes, and expression of aquaporins in a sex-dimorphic manner. This results in sex-related differentiations in distribution and function of adipose tissue [45]. Testosterone facilitates hypertrophy of muscle tissue through a complex process. Firstly, it enhances the transcriptional activity (and possibly the accretion) of myonuclei, increasing thereby the synthesis of contractile proteins. Secondly, it activates satellite cells, which in turn lead to the formation of new myotubes [46].

Body fat, muscle mass, and insulin sensitivity are physiologically interrelated. Adipose tissue modulates the resistance to insulin through the action of multiple mediators, including adipokines, pro-inflammatory and anti-inflammatory cytokines, and chemokines [47]. Skeletal muscle favors insulin sensitivity through glucose consumption for energy expenditure [48]. Therefore, cross-sex hormonal treatment may indirectly influence insulin resistance via changes in body fat mass and in skeletal muscle size. However, the existing evidence is not robust and further research is needed to investigate the relationships between body fat distribution, muscle mass, and insulin resistance, specifically in transgender people under hormonal therapy.

The direct effects of testosterone on insulin resistance are uncertain. In cisgender men with both hypogonadism and either prediabetes or diabetes mellitus, the long-term administration of testosterone may improve glycemic control and other components of metabolic syndrome [49,50]. Figure 1 presents possible effects of testosterone replacement treatment. The responsible biological mechanisms are largely unknown. It is possible that testosterone promotes the commitment of mesenchymal pluripotent stem cells into the myogenic lineage and inhibits their differentiation into adipocytes. Furthermore, it may affect metabolic processes in mature adipocytes and myocytes by modulating the expression of glucose-transporter type 4 and insulin receptor, and by acting upon the function of enzymes involved in glycolysis. Testosterone may also protect pancreatic beta-cells against glucotoxicity-induced apoptosis [51]. In contrast, hyperandrogenism in cisgender women with polycystic ovary syndrome is associated with increased insulin resistance [52]. Hence, chromosomal factors may also play a role which further implies that testosterone replacement treatment may act differently in transgender males and hypogonadal cisgender men.

Estrogen therapy is associated with an aggravated (or at least non-beneficial) impact on insulin resistance in transgender females. This is in accordance with the presence of increased insulin resistance in other hyperestrogenic states such as in oral contraceptive therapy with high doses of estradiol [53], polycystic ovary syndrome [54], and gestational diabetes [55]. On the other hand, insulin sensitivity diminishes in women after menopause [56]. Perhaps endogenous estrogens within a physiological window have a beneficial role in the regulation of insulin sensitivity, whilst pharmacological administration or excessive levels of estradiol have the opposite effect [36].

Ovary-derived estradiol could benefit glucose homeostasis through various mechanisms. Direct binding to estrogen receptors in pancreatic islets might enhance the production of insulin. Furthermore, endogenous estradiol is thought to have a positive effect on insulin signaling in insulin-sensitive tissues and on glucose uptake by the glucose-transporter type 4 in the skeletal muscle. In addition, it may act indirectly by regulating oxidative stress and reducing inflammatory states or by modulating other hormones (glucocorticoids, growth hormone) which in turn influence insulin secretion or action [57,58].

On the other hand, extragonadal or excessive circulating estradiol is associated with altered glucose metabolism. In cases of exogenous treatment or hyperestrogenemic syndromes, estradiol may interfere with the binding of insulin to its receptor, thereby inducing resistance to its action [59]. In addition, administration of replacement therapy or endogenous augmentation to supraphysiological levels may invoke abnormal hyperinsulinemia either via overstimulation of estrogen receptors in pancreatic beta-cells [60] or because of suppression of glucose-transporter type 4 in the skeletal muscle [61]. Figure 2 depicts the effect of exogenous estrogen on glucose homeostasis. Chromosomal factors may also differentiate the response of glucose metabolism to estradiol. However, the existing evidence is scarce and additional studies with a special focus on transgender recipients of estrogen are certainly needed to clarify the interplay of inherent sex, feminizing treatment, and glucose metabolism.

The prolonged subnormal cellular response to endogenous insulin may lead to overt diabetes mellitus. Furthermore, insulin resistance is associated with a series of adverse metabolic conditions, including hypertension, obesity, and dyslipidemia. The implications for the cardiovascular health of the recipients of cross-sex hormonal therapy in the long-term have not yet been determined. Therefore, targeted screening of the transgender population for markers of insulin resistance and diabetes may be necessary. Moreover, further research is needed to investigate the role of exogenous sex steroids in glucose homeostasis in trans people so the findings can guide clinical practice.

### 4.4. Treatment of Trans Individuals with Diabetes Mellitus

The pathogenetic hallmark of type 1 diabetes is primarily an autoimmune beta-cell deficiency. Therefore, treatment with exogenous insulin is undoubtably necessary for trans patients with type 1 diabetes. There are no specific guidelines for treating transgender adults with type 2 diabetes. It is reasonable that general recommendations about healthy lifestyle behaviors and self-management are essential. A patient-centered approach should guide the choice of pharmacological treatment. Individualized factors that affect the selection of the appropriate regimen include glycemic control, body weight, tendency to hypoglycemia, underlying comorbidities, safety and tolerability, cost and accessibility, and personal preferences. Every treatment should be regularly monitored for evaluating its efficacy and recording potential side effects and burden [62].

In case of recipients of cross-sex hormonal therapy, anti-diabetic agents with insulin-sensitizing properties should be considered. In this regard, biguanides and thiazolidinediones are apparent possible choices. Metformin is a biguanide that decreases hepatic gluconeogenesis, facilitates peripheral glucose uptake, and hampers glucose absorption by the intestine. Treatment with metformin is characterized by lack of weight gain and minimal risk of hypoglycemia. It is primarily used as a first-line medication provided that adequate liver and renal function are ensured and gastrointestinal side effects (nausea and diarrhea) do not occur [62,63]. Thiazolidinediones act through the interaction with peroxisome proliferator-activated receptors. They can decelerate the loss of function of beta-cells during the clinical progression of type 2 diabetes. However, safety concerns are related to heightened risk of myocardial infarction (with rosiglitazone), edema, weight gain, heart failure, and bone fracture [64].

Type 2 diabetes is usually a progressive disease and hence, many patients, including those following gender-affirming treatment, require a combination therapy in order to maintain their glycemic targets. Incretin-based medications enhance glucose-stimulated insulin secretion, slow gastric emptying, and suppress secretion of glucagon. Their use is associated with reduction in body weight, whilst it does not carry a risk of hypoglycemia [65]. Dipeptidyl peptidase-4 inhibitors are a class of anti-hyperglycemic medications which act on a similar pathway. They slow the degradation of endogenous incretins, mainly the glucagon-like peptide-1 and the gastric inhibitory peptide, by blocking the responsible enzyme [66]. Both categories of anti-diabetic agents are a stepwise choice for second-line therapy or even monotherapy in cases of intolerance or contraindication to metformin.

Sodium/glucose co-transporter-2 inhibitors reduce hyperglycemia by excreting excessive glucose in the urine, whilst alpha-glucosidase inhibitors delay the digestion of complex carbohydrates by inhibiting their enzymatic breakdown [67]. Thus, both of these classes of medications can indirectly decrease insulin resistance by lowering blood glucose. Regimens that provide cardiorenal protection should be preferred for patients with high risk of atherosclerotic disease, heart failure, or chronic kidney disease. The addition of insulin should be decided mainly if there is marked hyperglycemia, combination therapy fails, or elements of ongoing catabolism (weight loss, hypertriglyceridemia, ketosis) exist.

The social determinants of health should always be considered when treating individuals with dysglycemia. The cost of diabetes care has increased dramatically over the past years, creating problems in accessibility and affordability for patients and their families [68]. Unfortunately, the transgender population is commonly marginalized within the social fabric. Barriers to job opportunities (and consequently the lack of social insurance) further restrict trans people’s access to health care, especially in cases of low personal income [69]. Furthermore, transgender persons often encounter disrespect and discrimination within health care settings [70]. The anticipated mistreatment may lead trans patients to avoid health care [71], which can worsen hyperglycemia and facilitate the occurrence of complications in patients with diabetes.

## 5. Conclusions

Gender-affirming hormonal treatment is an essential means of improving the social and emotional well-being of transgender individuals [72]. In general, it is regarded as a safe intervention as long as recipients are appropriately monitored. Nevertheless, concerns have been raised about the effect of hormonal cross-sex therapy on glucose homeostasis, and consequentially, on microvascular structural integrity, especially among transgender females. It appears that trans women have higher rates of coronary artery and cerebrovascular disease, although existing studies are limited by the absence of large research populations and inadequate data acquisition [73]. It is not known whether worsening insulin resistance because of feminizing treatment contributes to the increased rates of major cardiovascular events among transgender women. Finally, there is a lack of compelling evidence that links gender-affirming treatment with testosterone to increased rates of cardiovascular disease among trans men.

It is plausible that cross-sex hormonal therapy can increase the risk of diabetes mellitus either via negatively influencing physiological mechanisms of insulin resistance or indirectly by increasing body mass index and altering body fat distribution. It is also possible that separate factors, other than gender-affirming medical treatment, are related to the development of diabetes. Transgender females may face gender minority stress that could fuel psychological stress and risky behaviors (such as smoking, alcohol abuse, unhealthy diet, and decreased physical activity). Additionally, trans individuals usually have less access to preventive health care services and thus, screening for and treatment of risk factors are delayed. Until stronger evidence is available, prevention and diagnosis of diabetes and cardiovascular disease in transgender people should follow established standards of care for the cisgender population [74].

If diabetes mellitus eventually occurs after the initiation of hormonal treatment, unique challenges emerge in clinical practice. Initially, the effects of exogenously administered sex steroids on the clinical course and severity of diabetes and its potential complications are not clear. The lack of specific guidelines about managing hyperglycemia in trans individuals creates uncertainty about the selection of appropriate anti-diabetic regimens. Secondly, the psycho-social impact of simultaneously identifying as transgender and suffering from diabetes mellitus could burden the emotional health and social integration of patients. For this reason, a frequent schedule for screening distress is necessary [75].

In conclusion, the safety of gender-affirming therapy for transgender individuals is an important concern in clinical practice. This review article sought to provide an insight to the effects of cross-sex hormonal treatment on glycemia and the association of sex steroids with the risk of diabetes mellitus. Existing knowledge on this area is limited due to the shortage of large-cohort studies. Although modest in content, current research supports the view that estrogen therapy reduces insulin sensitivity. The effects of testosterone administration are unclear. The most important measures to minimize the risk of dysglycemia and diabetes mellitus are the education of the recipients of sex hormones on lifestyle modifications to reduce the risk of insulin resistance and the appropriate clinical supervision by their health care providers [76]. Given the diverse medical needs of trans people, health care providers need to secure the safety of therapeutic interventions in this sensitive population.

## Figures and Tables

**Figure 1 biomedicines-11-00670-f001:**
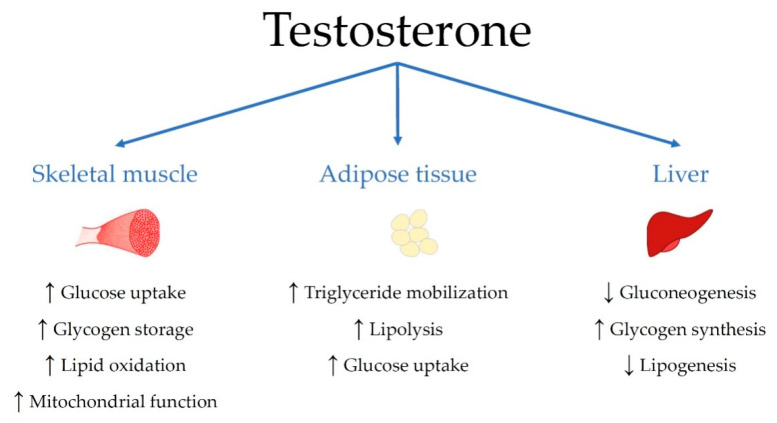
Potential actions of testosterone treatment on insulin-sensitive tissues.

**Figure 2 biomedicines-11-00670-f002:**
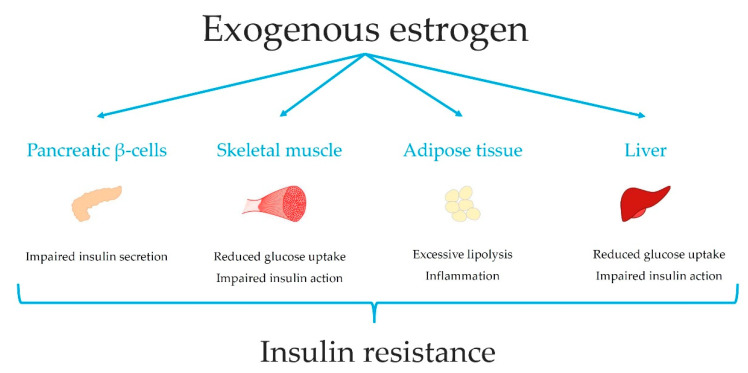
Estradiol-induced insulin resistance.

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
