# Peer review of "Glucose Homeostasis, Diabetes Mellitus, and Gender-Affirming Treatment"

_biomedicines, 2023, doi:10.3390/biomedicines11030670_

Round 1

Reviewer 1 Report

The authors submitted a narrative review in which they elucidated an effect of transgender treatment on glucose tolerance. The aim of the study is clear. The manuscript has a logical structure. The subsections of the aricles are well organased and contain comprehensibly reported innovative results. However, I would like to put forward several comments to dicsuss.

1. The authors should add tables with comparissons of designs, structures, results of the studies to easily understand a link between initial hypothesis and data, which confirmed it.

2. It would be great to give more representative explanations of underlying molecular mechanisms by which different kinds of therapies including hormonal care intervined in glucose homeostasis.

3. Clear schema or figura on issue mentioned above are welcome.

Author Response

We would like to thank the reviewers for their time and effort they spent to review our article. All of their suggestions were insightful and assisted us in improving the content of the text.

Rev 1

The authors submitted a narrative review in which they elucidated an effect of transgender treatment on glucose tolerance. The aim of the study is clear. The manuscript has a logical structure. The subsections of the articles are well organized and contain comprehensibly reported innovative results. However, I would like to put forward several comments to discuss.

  1. The authors should add tables with comparisons of designs, structures, results of the studies to easily understand a link between initial hypothesis and data, which confirmed it.

Two relevant tables were added in the paper.

  1. It would be great to give more representative explanations of underlying molecular mechanisms by which different kinds of therapies including hormonal care intervened in glucose homeostasis.

Further explanations about molecular mechanisms of action of sex steroids are provided in the revised text.

  1. Clear schema or figure on issue mentioned above are welcome.

Relevant figures were added.

Reviewer 2 Report

The review by Milionis et al. represents an excellent, very well-written, and informative summary of the (limited) evidence on the relationship between gender-affirming treatment and diabetes, a topic that is rarely discussed in the literature. 

I only have some minor comments and suggestions:

1.     Section 1, last paragraph: I would add to the important pleiotropic actions of insulin the anti-inflammatory ones (https://doi.org/10.1097/mco.0b013e3281e38774)

2.     Section 2, last paragraph: Although the available evidence is still inconclusive, I think that the risk of development of sex hormone-dependent malignancies should be mentioned (https://doi.org/10.1111/cen.13835).

3.     Section 3 and throughout the text: I would avoid using the term diabetic (“non-diabetic women”), as it is considered a stigmatizing language. The term “women or men with diabetes” seems more appropriate

4.     Section 4.1: Please elaborate on the reasons for the conflictive findings between the various studies.

5.     Section 4.4, second paragraph, line 2: Please remove the second appearance of the phrase 'insulin-sensitizing properties' (typo)

6.     Section 4.4: Evidence for cardioprotection with metformin is mainly observational. Furthermore, its use as a first-line treatment in all individuals with diabetes is no longer in accordance with current guidelines for the treatment of type 2 diabetes (https://doi.org/10.2337/dci22-0034).

7.     Section 4, paragraph: Sex hormones are involved in the different therapeutic responses of men and women to GLP-1 receptor agonists (https://doi.org/10.3390/jpm12030454)

8.     I would advise adding a figure summarizing the effects of sex hormones on glucose homeostasis.

Author Response

We would like to thank the reviewers for their time and effort they spent to review our article. All of their suggestions were insightful and assisted us in improving the content of the text.

Rev 2

The review by Milionis et al. represents an excellent, very well-written, and informative summary of the (limited) evidence on the relationship between gender-affirming treatment and diabetes, a topic that is rarely discussed in the literature.

I only have some minor comments and suggestions:

  1. Section 1, last paragraph: I would add to the important pleiotropic actions of insulin the anti-inflammatory ones (https://doi.org/10.1097/mco.0b013e3281e38774).

An addition to the text underlines the anti-inflammatory actions of insulin.

  1. Section 2, last paragraph: Although the available evidence is still inconclusive, I think that the risk of development of sex hormone-dependent malignancies should be mentioned (https://doi.org/10.1111/cen.13835).

A relevant reference was added to the text.

  1. Section 3 and throughout the text: I would avoid using the term diabetic (“non-diabetic women”), as it is considered a stigmatizing language. The term “women or men with diabetes” seems more appropriate.

The authors agree. The text was accordingly corrected.

  1. Section 4.1: Please elaborate on the reasons for the conflictive findings between the various studies.

Hypotheses about conflicting results were added to the text.

  1. Section 4.4, second paragraph, line 2: Please remove the second appearance of the phrase 'insulin-sensitizing properties' (typo).

The ‘typo’ was corrected.

  1. Section 4.4: Evidence for cardioprotection with metformin is mainly observational. Furthermore, its use as a first-line treatment in all individuals with diabetes is no longer in accordance with current guidelines for the treatment of type 2 diabetes (https://doi.org/10.2337/dci22-0034).

The text was modified according to the reviewer’s suggestion.

  1. Section 4, paragraph: Sex hormones are involved in the different therapeutic responses of men and women to GLP-1 receptor agonists (https://doi.org/10.3390/jpm12030454).

We agree that this is an important remark. However, we did not elaborate further on this issue because we feel that this is out of the scope of our review.

  1. I would advise adding a figure summarizing the effects of sex hormones on glucose homeostasis.

Relevant figures were added.

Reviewer 3 Report

The work submitted to Biomedicines by Millionis et al., titled: "Glucose homeostasis, diabetes mellitus, and gender-affirming treatment" is an interesting review regarding a topic of interest as gender issues seemingly are becoming more pertinent in the public dialogue and scientific input is thus becoming more important. 

The review is well organized and structured nicely with good flow making it easy for the reader to follow. 

The reviewer would like to offer the following points for consideration by the authors aiming at improving the manuscript.

1. Using a PRISMA diagram to illustrate the search strategy along with the criteria would be informative and powerful.

2. An interesting aspect in the discussion would be the microbiome. As there seems to be a relationship between the microbiome and diabetes/glycemia handling it would be interesting to discuss this topic while also add to it the element of hormonal intervention that may occur sometimes in individuals transitioning. A potentially useful review regarding the first portion (ie: the relationship between diabetes and the microbiome is the following for the authors to consider: 

Sikalidis, A.K.; Maykish, A. The Gut Microbiome and Type 2 Diabetes Mellitus: Discussing A Complex Relationship. Biomedicines 2020, 8, 8. https://doi.org/10.3390/biomedicines8010008.

Good job overall.

Author Response

We would like to thank the reviewers for their time and effort they spent to review our article. All of their suggestions were insightful and assisted us in improving the content of the text.

Rev 3

The work submitted to Biomedicines by Milionis et al., titled: "Glucose homeostasis, diabetes mellitus, and gender-affirming treatment" is an interesting review regarding a topic of interest as gender issues seemingly are becoming more pertinent in the public dialogue and scientific input is thus becoming more important.

The review is well organized and structured nicely with good flow making it easy for the reader to follow.

The reviewer would like to offer the following points for consideration by the authors aiming at improving the manuscript.

  1. Using a PRISMA diagram to illustrate the search strategy along with the criteria would be informative and powerful.

As this is a narrative review, no systematic process was used to identify the provided studies. Therefore, the depiction of a PRISMA diagram is not feasible.

  1. An interesting aspect in the discussion would be the microbiome. As there seems to be a relationship between the microbiome and diabetes/glycemia handling it would be interesting to discuss this topic while also add to it the element of hormonal intervention that may occur sometimes in individuals transitioning. A potentially useful review regarding the first portion (i.e.: the relationship between diabetes and the microbiome is the following for the authors to consider:

Sikalidis, A.K.; Maykish, A. The Gut Microbiome and Type 2 Diabetes Mellitus: Discussing A Complex Relationship. Biomedicines 2020, 8, 8. https://doi.org/10.3390/biomedicines8010008.

Certain remarks in the text were added to underline the importance of the sex-specific microbiome and the effect of sex steroids on it.

Good job overall.

Round 2

Reviewer 3 Report

The authors have made a reasonable effort in addressing reviewer's comments. Proofreading is suggested.